# Research into the Regenerative Braking of an Electric Car in Urban Driving

**Dainis Berjoza** [1,*], **Vilnis Pirs** [1] **and Inara Jurgena** [2]

1    Institute of Motor Vehicles, Technical Faculty, Latvia University of Life Sciences and Technologies, 5 J.Cakstes Boulevard, LV-3001 Jelgava, Latvia
2    Institute of Business and Technologies, Faculty of Economics and Social Development, Latvia University of Life Sciences and Technologies, 18 Svetes Street, LV-3001 Jelgava, Latvia
*    Correspondence: dainis.berjoza@lbtu.lv; Tel.: +371-29-735-949

**Abstract:** As the use of fossil energy sources in transport declines, new technologies, e.g., electric vehicles, are being introduced. One of the advantages of electric vehicles in urban driving is the possibility to charge their batteries with regenerative energy during braking. For this reason, electric cars usually have a longer range per charge in urban driving than in non-urban driving. This research experimentally examined the regenerative braking of a converted Renault Clio electric car at different regenerative braking settings in the range of 0–100%. An original research methodology was developed for road tests in urban driving. The driving cycle included aggressive driving with rapid acceleration and braking. The road test was conducted in second and third gears, which are the usual gears for driving an electric car in a city. The highest regenerative braking efficiencies were achieved at a 100% setting, which in some replications reached 24% of the total electrical energy consumed; however, the 100% setting was too high from the perspective of comfortable driving of the electric car and contributed to a too significant increase in the braking force at the initial stages of braking.

**Keywords:** regenerative braking; electric vehicle; electrical energy consumption

## 1. Introduction

Based on trends in automobile production and exploitation in the world, the number of electric vehicles has tended to increase in the last decade. Automobile manufacturing companies are committed to reducing $CO_2$ emissions from new automobiles; therefore, there are plans to significantly increase the production of electric and hybrid automobiles over the next decade. One of the most important advantages of electric vehicles is that the kinetic energy of vehicle mass in the braking process can be converted into other forms of energy and stored in batteries or other energy storage devices.

Regenerative braking efficiencies could be different during braking, depending on the settings available for some electric vehicle models. An excessively high regenerative braking efficiency can ensure a high energy return to the batteries, yet it does not provide enough comfort for the driver due to the fast deceleration. Several research studies have been conducted on regenerative braking.

The energy regenerated could be stored in different ways, e.g., in batteries or supercapacitors. The charged supercapacitors are able to supply a lot of electrical energy in a short period, which is necessary during the acceleration of cars. In this way, the vehicle batteries could be relieved, and the energy regenerated can be reused. The regenerative braking current might reach 200 A during braking. The main advantage of supercapacitors, compared with traditional batteries, is their ability to accumulate a large amount of electrical energy in a short period. Supercapacitors can extend battery life and increase the efficiency factor of an electric car system. Research studies have been conducted on an electric truck with a curb weight of 10 t, in which the supercapacitor was the only source of stored energy. The distance covered by the truck without a load reached 20 km, while with a load it was

4 km. Such a truck was intended for use in closed areas and with frequent acceleration and stopping, at a speed not exceeding 40 km h$^{-1}$. The tests were conducted without a load and with a load of 70 t. The highest regenerative braking efficiency, 40.7%, was achieved when operated without a load and 8.8% with a load [1].

Fiori C. et al. have developed a mathematical model of energy distribution and consumption for an electric automobile. The efficiency factor for energy regenerated is calculated as the ratio of actually regenerated energy to the theoretically possible one. The prototype of an electric car Nissan Leaf was road tested in six different driving cycles to validate the mathematical model. The research by C. Fiori et al. compared the electric car with an internal combustion engine car of a similar class, as well as electric vehicles with each other. The research found that the electric car consumed 82.5% less energy than an internal combustion engine car [2].

Energy regeneration is also possible for hydrostatic drive electric vehicles. In such vehicles, the energy regenerated could be stored in both electric batteries and hydraulic accumulators. According to the simulation data for hydrostatic drive vehicles, a higher regenerative braking efficiency, 43.35%, was achieved when storing energy in the battery than in the hydraulic accumulator [3].

Regenerative braking could also be used in cars running on hydrogen or equipped with hydrogen cells. Such automobiles are similar to traditional electric vehicles in terms of engine type and design, and only the technology of generating electrical energy is different, as it is generated in hydrogen cells. Research has been done on a hydrogen cell city bus that stored the electricity generated during braking in ultracapacitors, which were used along with conventional batteries. A strategy for regenerative braking coordination, which involves the front wheel, rear wheel and energy regeneration systems, has been developed by scientists. Braking tests have been performed on a low-adhesion road surface (coefficient of adhesion of 0.15). The regenerative braking control system developed allows 16% fuel savings. An electric bus with conventional batteries was also tested by performing a simulation by means of MATLAB/Simulink software. After the simulation, the bus was road tested based on the results of the simulation, and the energy regeneration system operating in conjunction with the ABS system was optimized [4,5].

Björnsson L.-H. and Karlsson S. have developed a mathematical model for the evaluation of potential regenerative braking energy. Road tests were carried out in real road conditions in Sweden, saving data in a data logger at a rate of 2.5 measurements per second. Average energy savings for a BEV (battery electric vehicle) reached 15%, while for a light hybrid vehicle it was 10%. Scientists have developed and tested a simplified power-based model for electric vehicles. The model developed was calibrated based on Nissan Leaf data obtained from road tests. The model identified energy savings during regenerative braking in different driving cycles. The highest efficiency, 31%, was achieved during a UDDS driving cycle. A simulation of the consumption of electrical energy by other systems, e.g., an air conditioner, revealed that the total effect was lower, and the regenerative braking efficiency reached only 21.3%. The model could also be used to simulate and identify a maximum range per charge for an electric car [6].

For an electric car with wheel motors to have high-quality electro-hydraulic regenerative braking, a mathematical algorithm that considers the height of the automobile's centre of gravity and the distribution of weight on the axles during braking has been developed. The brake force distribution algorithm can reduce rear axle regenerative braking during intensive braking [7]. Front-wheel-drive automobiles could also experience the loss of stability when braking on a low-adhesion road surface if the rear axle deviates from the intended trajectory.

Zhang J. et al. have succeeded in achieving a 25.7% efficiency by developing an electronic system of friction and regenerative braking. The system represents an electronic ABS module of modernized design that has replaced a standard ABS module. The module is connected with a motor inverter and functions according to the parameters of the motor and the braking mode. The moment that the driver takes the foot off the accelerator

pedal, just like in an internal combustion car, the electronic ABS module starts regenerative braking with a low torque equal to that of the motor. The road test has recorded the electric car's speed and deceleration, as well as fluid pressure in the brake system and regenerative braking torques and currents. When braking is started, the regenerative braking torque increases in proportion to the pressure applied to the brake pedal. If the regenerative braking torque is not sufficient (at a certain brake pedal position), the conventional braking system is engaged. The road tests were replicated also in the ECE driving cycle [8].

Regenerative braking could be applied in case of multi-motor drive, e.g., if the motors are placed in the wheels. In this case, a special communication and control system is required, which ensures an accurate distribution of regenerative braking torque between the wheels. The system stores electrical energy both in conventional electric batteries and in supercapacitors. Researchers have developed a mathematical model for a four-motor electric car and simulated braking at low, medium and high braking intensity [9].

Various research studies have focused on regenerative braking performed in various conditions by applying the electric braking method. A special test bench has been developed for performing such tests. The test bench allows the main parameters of an electrically adjustable braking system to be simulated: braking time, braking torque and energy regenerated. It also enables the brake pedal to be set in four different positions, thereby providing different braking modes [10,11].

An electric car can regenerate energy not only during braking but also when driven on rough roads due to the vibrations of the automobile. Such an idea arose because an electric car with in-wheel motors is less controllable and provides less driving comfort on bumpy roads than that with conventional motor placement. The main cause of this deterioration is an increase in the unspring weight of the automobile. To regenerate energy from the vibrations of the car's body, special shock absorbers have been developed, which generate electricity by means of the electromagnetic field during the movement. Such a system not only generates energy but also improves driving comfort [12].

Kullingsjö L.-H. and Karlsson S. have developed a computational model for electric and hybrid cars to identify total national energy savings from regenerative braking. In this research, an automobile weighing 1500 kg was chosen as the most common automobile operated in the country. The research analysed 430 automobiles equipped with a GPS logger over a period of two months. The age of the automobiles was less than 9 years. In total, the automobiles had travelled 1.14 million km. Energy saving potential for battery electric vehicles (BEVs) and hybrid electric vehicles (HEVs) was estimated based on real-life driving habits. An economic effect was calculated based on the average distance covered, which on average reached 330 EUR per year for each vehicle [13].

Regenerative braking has been optimized for other vehicles as well, such as an electric train. As a result of the simulation, significant energy saving for the electric train was obtained, reaching 25.0% [14].

Many studies focus on modelling the energy regeneration process and improving the algorithm; however, the regeneration process has been researched relatively little in real driving conditions on specific routes. Accordingly, the present research aims to examine the energy regeneration process of a converted electric car in city driving conditions at different regenerative braking settings.

## 2. Materials and Methods

The aim of the road test was to identify the amount of energy regenerated in a moderately aggressive driving style at different settings of regenerative braking. The road test identified the amount of energy technologically regenerated by the mentioned electric car, as well as a subjective rating of braking effectiveness in terms of ergonomics and possibility to control the braking force. A converted Renault Clio electric car was used in the road test. Before the conversion, the electric car was equipped with a 1.2 L gasoline engine. The main technical characteristics of the electric car are presented in Table 1.

**Table 1.** Main technical characteristics of the converted electric car Renault Clio.

| No | Parameter Characteristics | Parameter Value |
|---|---|---|
| 1. | vehicle category | M1 |
| 2. | motor nominal power | 30 kW |
| 3. | controller type | Sigma Drive PAC950TL02 |
| 4. | gear box | 5 speed manual |
| 5. | gear ratios | |
| | 1 | 3.363 |
| | 2 | 1.864 |
| | 3 | 1.321 |
| | 4 | 1.029 |
| | 5 | 0.821 |
| | final drive | 4.067 |
| 6. | maximum speed | 120 km h$^{-1}$ |
| 7. | battery cells | LiFePO4, 32pcs. |
| 8. | total battery voltage | 102.4 V |
| 9. | on-board energy | 10.5 kWh |
| 10. | nominal voltage of a battery cell | 3.2 V |
| 11. | voltages set in the BMS | $U_{min}$ = 2.6 V; $U_{max}$ = 3.8 V |
| 12. | range per charge | 60 km |
| 13. | minimum BMS response time | 0.015 s |
| 14. | battery charge time | 3.5 h |
| 15. | battery capacity | 100 Ah |
| 16. | max. peak discharge current | <1000 A |
| 17. | max. peak discharge current | <300 A |
| 18. | max. charge current | <300 A |
| 19. | curb weight | 1080 kg |
| 20. | weight in the road test | 1230 kg |

An original research methodology was developed for road tests. The highest regenerative braking efficiency of an electric car is typical in urban driving, as the regenerative braking can provide a longer range per charge, compared with non-urban driving. A special city route was developed for the road test. The route involved main driving regimes in urban driving, i.e., movement in a yard, and movement through four intersections, two of which were controlled by traffic lights. The route had a 60 m road section with a speed limit of 30 km h$^{-1}$, which included an unregulated T-junction with three pedestrian crossings. Approximately 250 m of the route included main streets. The road test route is shown in Figure 1.

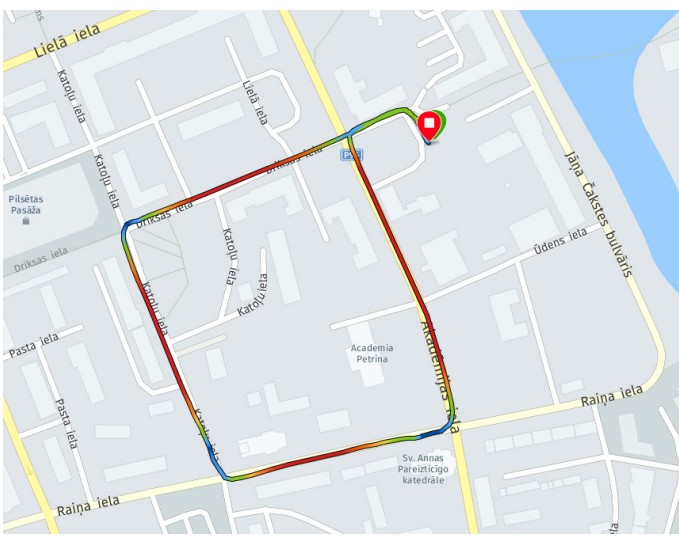

**Figure 1.** Road test route in Jelgava city.

A sample of data recorded by a Garmin EDGE 830 speed logger is presented in Figure 2. The length of the road test route was 1.06 km (1.06 km travelled on the specific route $v_{avg}$ = 19.4 km h$^{-1}$, road test duration t = 3:16 min).

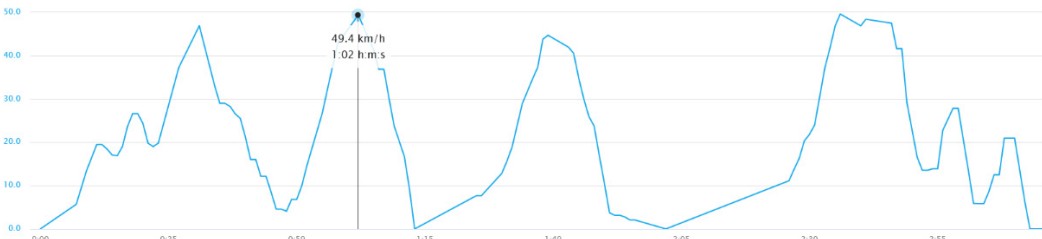

**Figure 2.** Changes in speed on the road test route.

A data logger Graphtec midi Logger GL220 (Dataq Instruments Inc., Akron, OH, US) was used to record electrical data. The logger allowed us to set a rate of measurements from 10 times per second to once per 24 h. The device was equipped with a 4.3-inch screen. A voltage between 8.5–24 V could be used to power the logger. The data logger had 10 analogue input channels. Electrical connections of test equipment is shown in Figure 3.

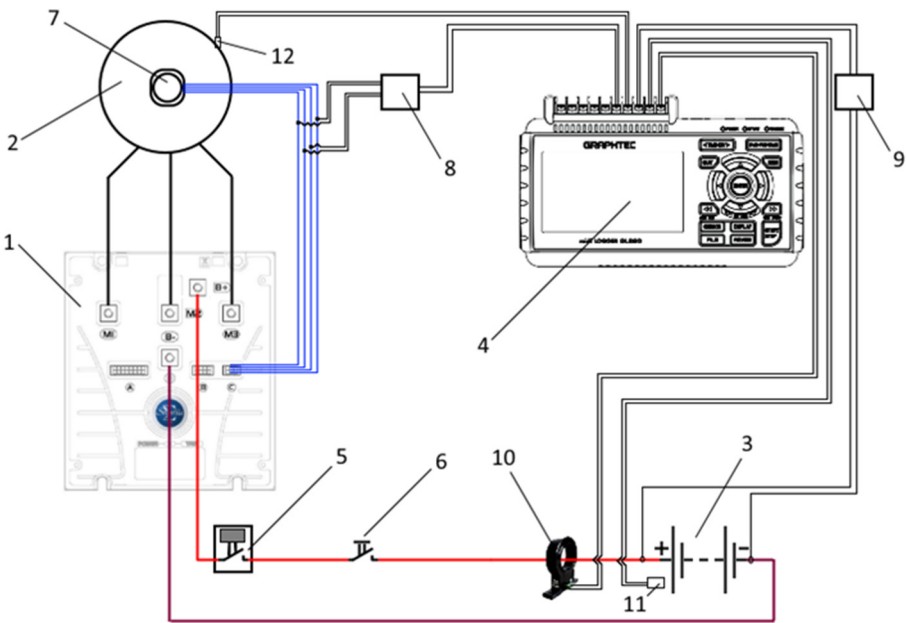

**Figure 3.** Electrical connections of test equipment: 1—Sigma Drive controller; 2—electric motor; 3—96 V battery pack; 4—data logger Graphtec GL220; 5—electromagnetic safety switch; 6—manual safety switch; 7—speed sensor; 8—frequency converter; 9—voltage sensor; 10—current sensor; 11 and 12—temperature sensors.

The road test was carried out at an ambient temperature of +10–+18 °C in Jelgava city on 5 October 2021–3 November 2021. The road test was performed by two operators. One of the operators drove the car, while the other was responsible for starting the route and electrical parameter loggers and stopping them at the end of the road test. It was possible for the controller of the electric car to make different settings for regenerative braking, which during the road test was set in the range of 0–100% with an increment of 20%. In this way, the road test was carried out at 6 settings. The road test was started at a 0% setting, then 20%, 40%, etc. For the battery charge level not to affect the road test, a setting was made before each road test replication. The road test had 10 replications to exclude the influence of different traffic intensity. The road test was carried out in the period from 10.00 to 13.00 to have similar traffic intensity. The road test was performed in 2nd and 3rd gears.

LED daytime running lights were used for the car in the experiment, which were powered by a 12 V battery. The impact of lamps and on-board system consumers on the overall energy balance was negligible and the consumption did not exceed 150 W.

When the car was ready for the road test, both loggers were turned on and movement was started. The road test represented a moderately aggressive driving style, seeking to reach and maintain the maximum permitted speed at the points of the route. When the electric car was stopped after completing the route, the second operator stopped both loggers and saved the data. In parallel with saving data, other details were also recorded: the selected driving regime, the selected gear, the number of the road test replications, etc. The road test was carried out with fully charged batteries until their remaining energy was not less than 40%, which was tentatively controlled by the battery charge indicator in the on-board system of the electric car. After the road test, the data were transferred to the computer and processed.

For the converted electric car, the Sigma Drive controller provided only one programming option to start energy regeneration. The energy regeneration was started when the brake pedal was pressed and the stop lights lit up. For this reason, to reach the highest possible regenerative braking efficiency in the road test, the braking was started by pressing the brake pedal as slowly as possible so that the regeneration began, yet the main or friction braking system would not start working.

Activation of regenerative braking for the converted electric car is different from strategies of regeneration of most industrially produced electric cars. The intensity of braking for the converted electric car has been programmed using the Chassis Dynamometer.

The regeneration braking parameters for the experimental car, similar to also the driving parameters, are programmed in the Sigma Drive controller at four different points of the electromotor rotation speed. These speed points are called: Motor Speed Minimum, Motor Speed Boost, Motor Speed Base and Motor Speed Maximum. In the process of the electric car development, the optimal brake slip and brake voltage settings were experimentally determined for every of the above mentioned speed points. These settings determine the character of the car regeneration braking at any possible car movement speeds or any electromotor rotation frequency. The parameters influencing braking are selected, balancing the driving comfort and the regeneration efficiency in the process of braking. During exploitation, regeneration, and with this also the intensity of braking the car, can be made more or less efficient by changing a parameter such as braking percentage levels. It is possible to change this parameter in the range from 0–100%. At the value of 0%, there is practically no regeneration in the process of braking; in turn, the value of 100% means that regeneration takes place with the maximal intensity in compliance with the programmed braking parameter values in the controller.

After the road test, data from 10 replications were selected for processing. The road test recorded ambient temperature, temperatures of the motor and the controller, battery voltage, battery charge or discharge current, as well as electric motor speed. After the road test, the speed of the electric car was calculated, as the transmission gear and motor speed were known:

$$v = \frac{\pi n_e r_k}{30 i_T},\tag{1}$$

where: $n_e$ is motor speed, min$^{-1}$; $r_k$ is the kinematic radius of the wheels of the electric car, m; and $i_T$ is the transmission gear ratio.

A transmission gear ratio is calculated according to the equation:

$$i_T = i_k i_0,\tag{2}$$

where: $i_k$ is the gear ratio of the gearbox and $i_0$ is the gear ratio of the final drive.

The kinematic radius of the wheels is calculated according to the equation:

$$r_k = 0.0127 d_r + 0.91 b_r k_r, \; m,\tag{3}$$

where: $d_r$ is the tire inner diameter in inches, $b_r$ is the tire width in meters and $k_r$ is the tire height factor.

Inserting Equations (2) and (3) into Equation (1) gives an equation for calculating the speed:

$$v = \frac{\pi n_e (0.0127 d_r + 0.91 b_r k_r)}{30 i_k i_0} \qquad (4)$$

The power consumed from the battery and regenerated is calculated according to the equation:

$$P = IU, \ W, \qquad (5)$$

where: $I$ is the current consumed from the battery or regenerated, A; and $U$ is the battery voltage, V.

Electrical energy consumed from the battery and regenerated is calculated according to the equation:

$$E = Pt = IUt, \ Ws, \qquad (6)$$

where: $t$ is measurement time, s.

During the road test, positive and negative currents were selected for the data to be recorded. The positive current showed the discharge of the battery, while the negative one showed the charge of the battery.

## 3. Results and Discussions

After the road test, the data were processed, and averages were calculated for different driving regimes and energy regeneration settings from 10 replications. The main data recorded and calculated during the road test and their changes are shown in Figure 4. The change in voltage was not significant and varied between 96.2 and 103.4 V. The change in current depended on the driving regime. The maxima of current were reached when accelerating the electric car. Since the road test route also had two traffic lights and five intersections, it was necessary to repeatedly decrease and increase speed, as well as stop at a red traffic light. For example, at the 108th second of the road test, the car stopped at a traffic light and started moving only at the 144rd second.

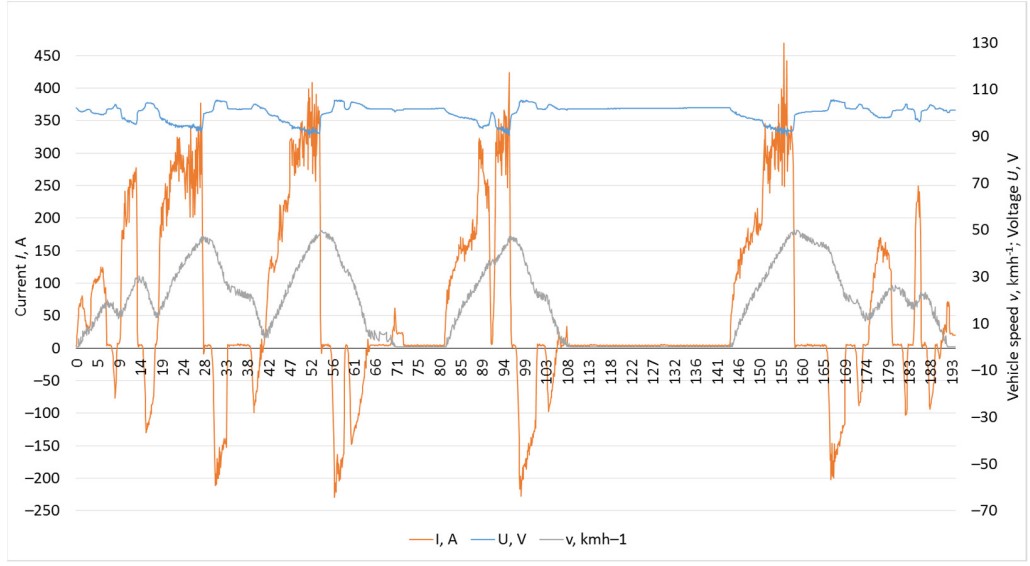

**Figure 4.** Example of road test data recorded when driving in 2nd gear at an 100% regenerative braking setting.

Since the road test was done in real traffic conditions and there were two traffic lights on the route, the test had 10 replications. The time spent on the route depended on the dynamic parameters of the car, e.g., the acceleration time and regenerative braking time. In

second gear at all regenerative braking settings, the time spent on the route was less than that in third gear and ranged from 184.5 s to 199.7 s (Figure 5). In third gear, the dispersion of the time spent on the route was greater and ranged from 201.0 s to 253.4 s. The largest difference in the time spent on the route was found at a 0% regenerative braking setting, which was 36.8% higher in third gear than in second gear. The smallest difference in the time spent on the route, 4.9%, was at a 100% regenerative braking setting, which could be due to the similar braking effectiveness and braking deceleration as the speed decreased. In second gear, the experimental car had better dynamic performance; therefore, it completed the road test route faster in all driving regimes.

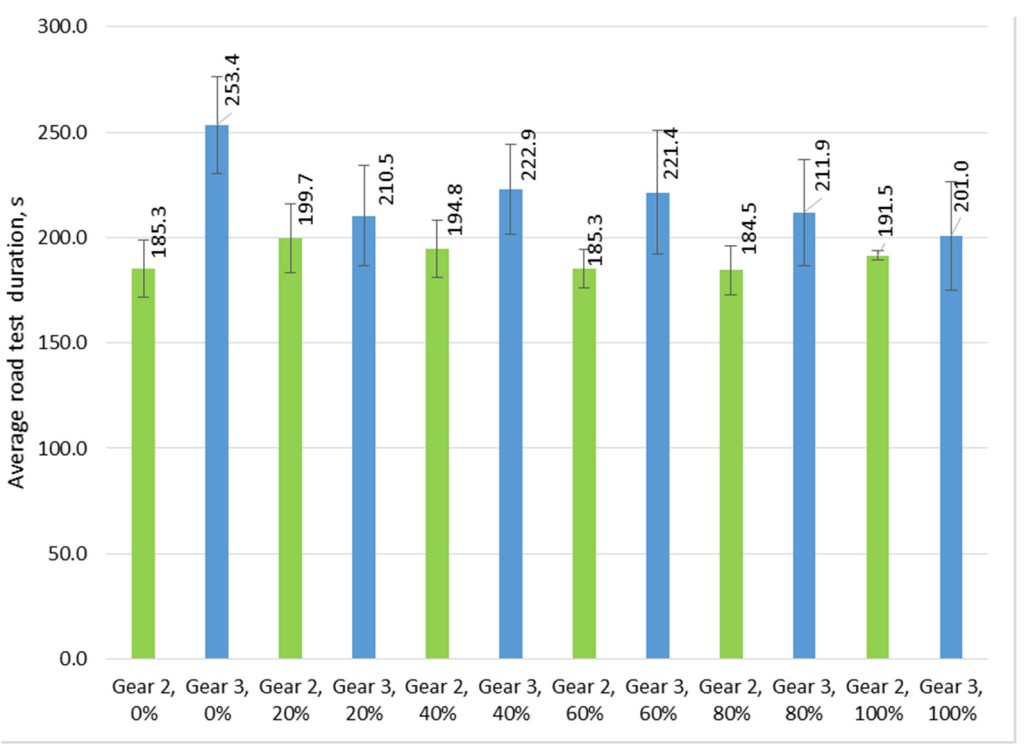

**Figure 5.** Average duration of road test replications at different regenerative braking settings.

One of the parameters indicating energy regeneration and efficiency is maximum current during regenerative braking, which is represented by negative values (Figure 6). During the road test, the maximum current was recorded in second gear at a 100% setting—230.6 A—while in third gear, it was 29.7% lower. A similar trend was observed for the other regenerative braking settings. At 0–40% settings, the regenerative current did not exceed 34.7 A and did not provide a significant energy return effect. A comparison of the energy regeneration process at a 60% and an 80% setting revealed a significant increase in current, almost two-fold, when driving in second gear, as the current increased from 78.1 A to 145.4 A. The maximum battery charging and discharging current during the regeneration process did not exceed the maximum permissible battery parameters (Table 1).

The basic idea of the energy regeneration process is to return the energy generated during braking to the battery of the electric car, as shown in Figure 7. Similar to the maximum regenerative braking current, the energy regenerated at settings up to 40% was not significant and did not exceed 17.00 Wh in second gear and 9.18 Wh in third gear on the road test route. In second gear at an 80% setting, an average of 82.32 Wh was returned to the battery, which was 42% more than at a 60% setting. Increasing a regenerative braking setting to 100% in second gear generated 92.34 Wh, which was only 12.2% more than at an 80% setting. A similar trend in efficiency increase at the highest settings was observed also in third gear, yet the absolute battery charging energy values were 29.5–43.4% lower than those in second gear at the settings from 60 to 100%.

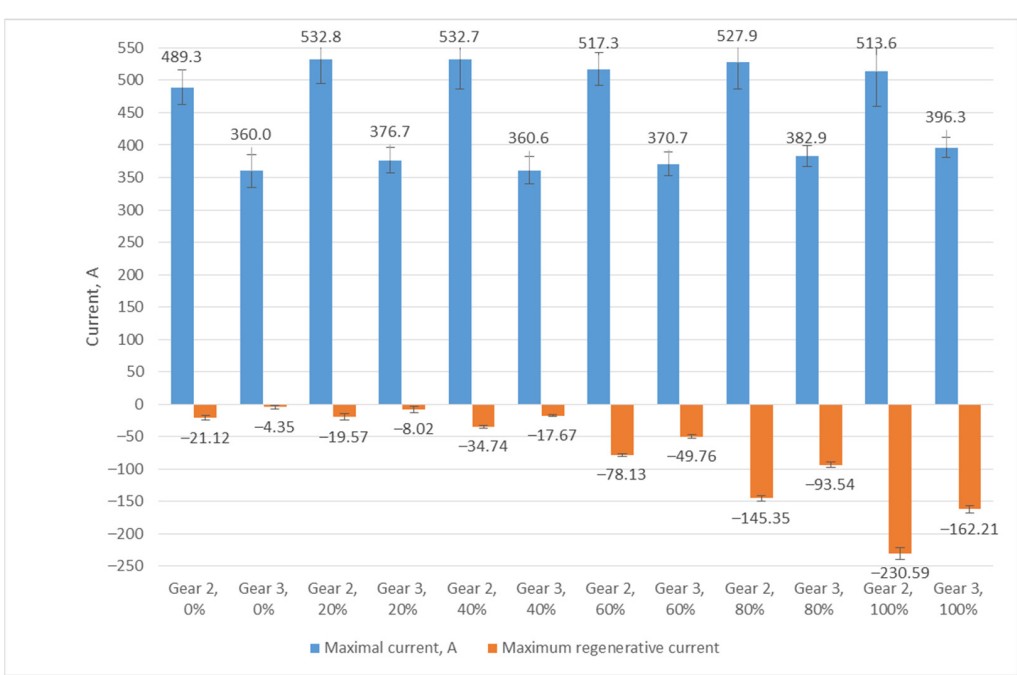

**Figure 6.** Maximum current when accelerating and applying regenerative braking.

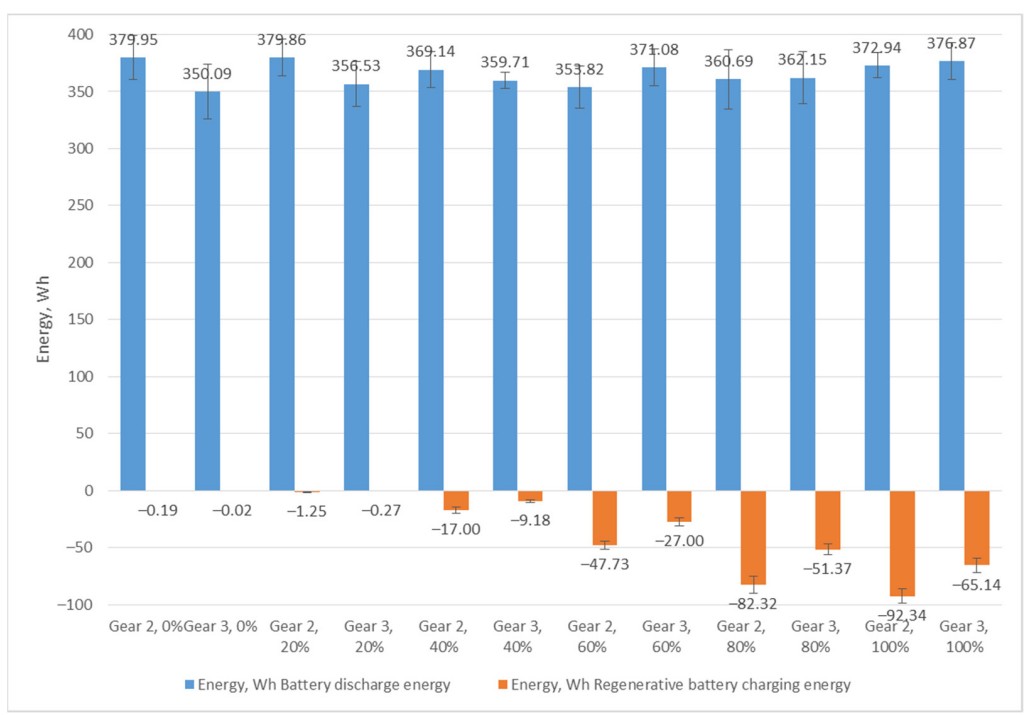

**Figure 7.** Battery discharge and regenerative braking charging energy during the driving cycle.

During the movement of the electric car, the energy balance could be divided into two parts: the energy consumed from the batteries during acceleration and cruising and the energy regenerated during braking, which is returned to the batteries. Figure 8 shows calculation results for the road test: the energy regenerated during regenerative braking as a percentage of the energy consumed for the movement of the car. At a 40% regenerative braking setting, the energy regenerated did not exceed 4.63% in second gear and 2.56% in third gear, which was a very low percentage and was not significant for increasing the driving range. The highest percentages of regenerated energy were achieved in 2nd gear at an 80% setting (22.84%) and at a 100% setting (24.74%). In third gear, which was used

mainly in non-urban driving, less energy was regenerated during regenerative braking. Given that at a 100% regenerative braking setting the braking of the experimental electric car by means of the motor was very fast and difficult to control by the force of pressing the brake pedal, it was useful to choose a setting that was within 70–80%. This range of settings could be individually customized depending on the driving habits and experience and the driving style of the electric car driver.

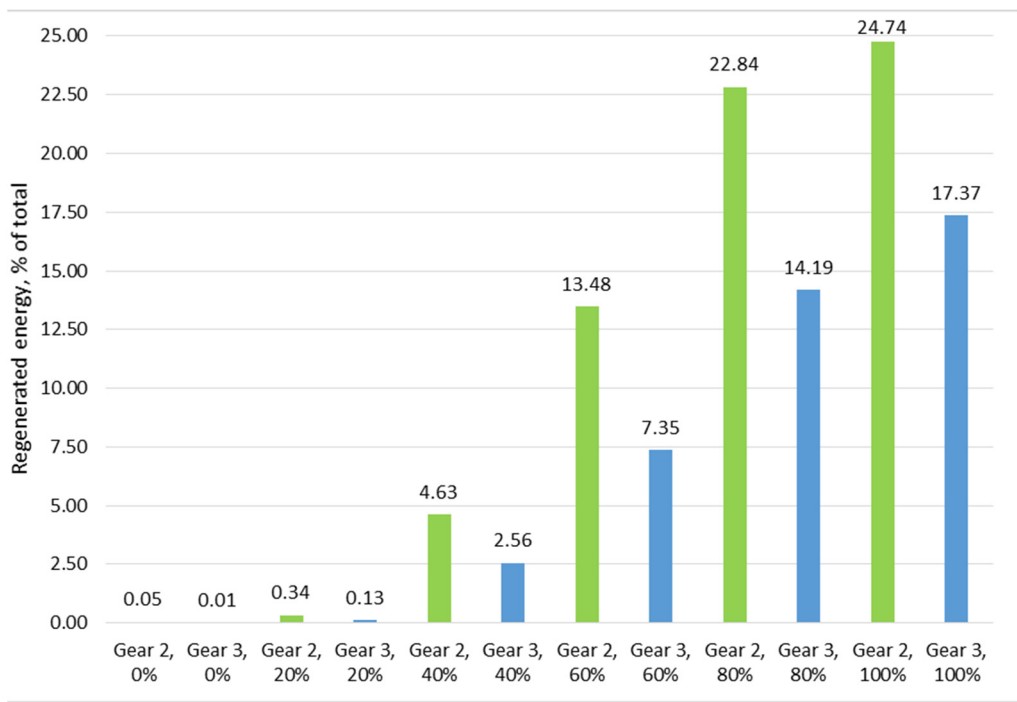

**Figure 8.** Regenerated energy as a % of total energy when driving in 2nd and 3rd gears at different regenerative braking settings.

At the previous stages of our research, the focus was placed on identifying the maximum regenerative braking efficiency for the converted Renault Clio electric car that had been accelerated up to 50 km h$^{-1}$ and braked to a complete stop by means of only regenerative braking [15]. In the road tests, the maximum efficiency was achieved in second gear, 32.5%, at an 80% regenerative braking setting and 35.1% at a 100% setting. In third gear, the figures were 23.5% and 25.4%, respectively. A comparison of the results of research on maximum regenerative braking efficiencies and the results achieved in road tests in real urban driving revealed that the efficiencies were on average 29–48% lower at regenerative braking settings in the range of 60–100%.

## 4. Conclusions

- An experimental route for research on energy regeneration in electric car has been designed and tested, which included various road sections with intense city traffic, involved several driving regimes and had regulated and unregulated intersections. The length of the route was 1.06 km, and the average time spent on the route was in the range of 185–253 s;
- Most of the available research studies focused on examining energy regeneration by employing a mathematical model or doing tests on a test bench and identified regenerative braking efficiencies in the range of 25.7–31%;
- The experimental electric car was tested in urban driving in the two most frequently used gears—second and third—at regenerative braking settings in the range of 0–100%. At settings above 80%, it was difficult to control the braking force because the energy regeneration started at the same time as the car's stop lights lit up;

- Due to better car movement performance in second gear, the time spent on the route at all regenerative braking settings was less by 5–37% than that in third gear;
- An analysis of maximum regenerative currents revealed that the highest current was recorded in second gear at a 100% setting—230.6 A—which was 29.7% lower than that achieved in third gear. At an 80% regenerative braking setting, the regenerative current was 37% lower in second gear and 42.4% lower in third gear than that at a 100 setting. At regenerative braking settings less than 60%, the regenerative current was relatively low;
- The energy consumed when driving in second gear varied from 353.8 to 379.95 Wh, and in third gear from 350.09 to 376.87 Wh. The higher energy consumption in second gear was offset by more energy regenerated during regenerative braking. During regenerative braking in second gear, 47.7 Wh at a 60% setting, 82.3 Wh at an 80% setting and 92.3 Wh at a 100% setting were supplied to charge the batteries. In third gear, the values were 27 Wh, 51.4 Wh and 65.1 Wh, respectively. At the lowest settings, the energy regenerated did not exceed 17 Wh, and the benefit of increasing the driving range of the electric car was insignificant;
- In real city driving conditions, the experimental electric car could achieve a regenerative braking efficiency of 24.7% in second gear at a 100% setting. In practice, a recommended setting should not exceed 80%, resulting in a regenerative braking efficiency of 22.8% in second gear and 14.2% in third gear;
- When making regenerative braking settings in an electric car, it is necessary to consider the maximum settings and have the stop lights lit up in case of fast decelerations to ensure traffic safety on the road. Special attention should be paid to a decrease in energy regeneration if regenerative braking is started automatically after releasing the accelerator pedal.

**Author Contributions:** Conceptualization, D.B. and I.J.; methodology, D.B. and V.P.; software, D.B. and V.P.; validation, D.B., I.J. and V.P.; formal analysis, D.B.; investigation, D.B. and I.J.; resources, D.B.; data curation, D.B. and V.P.; writing—original draft preparation, D.B.; writing—review and editing, D.B. and I.J. All authors have read and agreed to the published version of the manuscript.

**Funding:** This research received no external funding.

**Institutional Review Board Statement:** Not applicable.

**Informed Consent Statement:** Not applicable.

**Data Availability Statement:** The data presented in this study are available on request from the corresponding author.

**Conflicts of Interest:** The authors declare no conflict of interest.

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
