# Peer review of "Research into the Regenerative Braking of an Electric Car in Urban Driving"

_wevj, doi:10.3390/wevj13110202_

Round 1
Reviewer 1 Report
The timing and intensity of regenerative braking are affected by many factors, such as vehicle class and control strategy, etc. Simply setting the braking percentage is not in line with engineering practice. It is suggested to give the optimal braking strategy of the research object under the current road conditions from the comprehensive consideration of energy recovery efficiency and comfort (as mentioned in this paper).
Author Response
Dear reviewer!
The authors are grateful to you for the work done in reviewing our article and for the comments and suggestions that are given in your review.
The timing and intensity of regenerative braking are affected by many factors, such as vehicle class and control strategy, etc. Simply setting the braking percentage is not in line with engineering practice. It is suggested to give the optimal braking strategy of the research object under the current road conditions from the comprehensive consideration of energy recovery efficiency and comfort (as mentioned in this paper).
Sigma Drive PAC950TL02 controller has been used for the experimental electric car Renault Clio. This type of controller allows for making several parameter settings and changes during the configuration process of electric cars. For the regenerative braking efficiency setting, this controller allows for only one parameter change, making changes to the regenerative braking torque by changing the degree of regeneration expressed as a percentage. The controller used in the design of the converted car has simplified options for setting up the regeneration process, which may differ from the regeneration algorithms of modern industrially produced electric cars. The activation of the regeneration process is also different from standard electric cars. For example, on the Mitsubishi iMiEV electric car and other similar prototypes, regenerative braking is activated when you take your foot off the accelerator pedal. The main advantage of such an activation is that the regeneration is activated even if the brake pedal is not pressed. The disadvantage of this system is that the brake lights do not come on. This problem has been fixed in cars of later years. If the car's brake lights do not turn on during the regeneration process, the degree of regeneration should not be adjusted excessively high, because the car driving behind is not informed about the start of the braking process of the car in front, which can cause a traffic accident. Therefore, with such a regeneration system the activation algorithm recommends that the deceleration occurring during the regeneration of the car should be set no higher than it could be for similar internal combustion engine cars. The experimental prototype Renault Clio has only one option for activating regenerative braking, by connecting to the brake light activation switch. Therefore, in the current version of the rebuilt electric car, there is no possibility to change the braking algorithm in a simplified way. Undoubtedly, the kinetic energy accumulated by the car during the acceleration process depends on the mass of the car and the speed achieved. However, in the overall energy balance, a heavier car will consume more energy during the acceleration process, but more energy will be recovered during the braking process. Conversely, a lighter car will consume less energy during the start-up process and less energy will be regenerated during the braking process. It is possible that the percentage of the regenerative energy recovered will be analogous for electric cars of different masses at similar settings.
The methodology is supplemented with an explanation of the regeneration settings in the process of adjusting the electric car.

Reviewer 2 Report
The paper is interesting and discusses important subject of regenerative braking of electric vehicle. Some minor improvements could, however, improve quality of the paper:
1. In introduction, phrasing "nation + scientists" is frequently used. Instead, using phrase "First Author Name et al." would improve readability, avoiding confusion for the reader (for example, are Swedish scientists mentioned in line 77 the same as in line 129?).
2. Authors did not mention values of maximum charging and discharging current of the battery. Can it be assumed that currents during both acceleration and braking were within allowed bounds?
3. Similarly to 2., auxiliary systems of the vehicle (lighting, onboard computer etc.) were not mentioned. Did those have any impact on regenerative braking or were negligible (or powered from separate source)?
4. Authors could consider plotting voltage and velocity waveform on the second axis for better readability (Fig. 4).
Minor grammar check is also advisable.
Author Response
Dear reviewer!
The authors are grateful to you for the work done in reviewing our article and for the comments and suggestions that are given in your review.
- In introduction, phrasing "nation + scientists" is frequently used. Instead, using phrase "First Author Name et al." would improve readability, avoiding confusion for the reader (for example, are Swedish scientists mentioned in line 77 the same as in line 129?).
All references in the introduction, where the nationality of the authors is included, have been corrected according to the reviewer's recommendations.
- Authors did not mention values of maximum charging and discharging current of the battery. Can it be assumed that currents during both acceleration and braking were within allowed bounds?
The maximum battery charging and discharging current during the regeneration process did not exceed the maximum permissible battery parameters. (Included in the analysis of Figure 6) The Table 1 is supplemented with the maximum values of the battery charging and discharging current.
- Similarly to 2., auxiliary systems of the vehicle (lighting, on-board computer etc.) were not mentioned. Did those have any impact on regenerative braking or were negligible (or powered from separate source)?
LED daytime running lights are used for the car in the experiment, which are powered by a 12 V battery. The impact of lamps and on-board system consumers on the overall energy balance is negligible and the consumption does not exceed 150W. The explanation is included in the methodology of the experiments.
- Authors could consider plotting voltage and velocity waveform on the second axis for better readability (Fig. 4).
An additional axis has been added to the voltage and speed curves in Figure 4.

Reviewer 3 Report
This paper research into the regenerative braking of an electric car in real driving conditions on specific routes. Specifically, the research experimentally examined the regenerative braking of a converted electric car Renault Clio at different regenerative braking settings in the range of 0-100%. The highest regenerative braking efficiencies were achieved in the experiment at a 100% setting, which in some replications reached 24% of the total electrical energy consumed; however, the 100% setting was too high from the perspective of comfortable driving of the electric car and contributed to a too significant increase in the braking force at the initial stages of braking.
I have some main concerns regarding this paper.
---------------
1. Why is there still a small amount of regenerative energy when the regenerative braking settings is 0%.
2. In line 255, please explain why 2nd gear has better dynamic performance from the perspective of mathematical model.
3. In urban rail transit, Shuai Su from Beijing Jiaotong University and other scholars have been studying the impact of regenerative braking on train operation, and suggested reference, i.e. “An Energy-Efficient Train Operation Approach by Integrating the Metro Timetabling and Eco-Driving”, “An integrated energy-efficient train operation approach based on the space-time-speed network methodology”, “Energy-efficient operation by cooperative control among trains: A multi-agent reinforcement learning approach”.
The comments below are mainly editorial matters:
4. In equation (6), it is recommended to use the International System of Units (SI) for energy.
Author Response
Dear reviewer!
The authors are grateful to you for the work done in reviewing our article and for the comments and suggestions that are given in your review.
- Why is there still a small amount of regenerative energy when the regenerative braking settings is 0%.
At 0% regeneration rate, the regenerative system should theoretically be disabled. However, in practice, setting this mode does not completely turn off regeneration, as a mechanical switch would. Therefore, a 0% recovery rate is the smallest possible adjustable regeneration that is practically imperceptible. This parameter can be used for comparison as the output data for other adjustments.
- In line 255, please explain why 2nd gear has better dynamic performance from the perspective of mathematical model.
In 2nd gear, the car has a higher gear ratio than in 3rd gear. For this reason, in gear 2, at a constant engine torque, a greater torque is obtained at the wheels, which provides higher dynamism indicators of electric cars, for example, a shorter acceleration time to a certain speed and a higher potential acceleration.
- In urban rail transit, Shuai Su from Beijing Jiaotong University and other scholars have been studying the impact of regenerative braking on train operation, and suggested reference, i.e. “An Energy-Efficient Train Operation Approach by Integrating the Metro Timetabling and Eco-Driving”, “An integrated energy-efficient train operation approach based on the space-time-speed network methodology”, “Energy-efficient operation by cooperative control among trains: A multi-agent reinforcement learning approach”.
It is nice to know that significant energy savings can also be made for rail vehicles through energy regeneration during the braking process. The literature analysis was supplemented.
- In equation (6), it is recommended to use the International System of Units (SI) for energy.
The System of Units (SI) unit of energy is J. However, electrical energy is more often measured in Ws, Wh or kWh. Since the analysis of the article uses energy in Wh (Figure 7), we believe that it is not useful to use two different systems of electrical energy units in one article. Made minor corrections in the description of equation 6.

Round 2
Reviewer 1 Report
The current version is available for publication.